# Association of Tdap vaccine guidelines with vaccine uptake during pregnancy

**Julia D. DiTosto[1], Rebecca E. Weiss[2], Lynn M. Yee[1], Nevert Badreldin[1]***

**1** Division of Maternal-Fetal Medicine, Department of Obstetrics and Gynecology, Northwestern University Feinberg School of Medicine, Chicago, IL, United States of America, **2** Department of Pediatrics, Northwestern University Feinberg School of Medicine, Chicago, IL, United States of America

* nevert.badreldin@northwestern.edu

**Data Availability Statement:** Due to potentially identifying and sensitive patient information, we are unable to share our dataset. However, if someone is interested in requesting access to the data, they may contact the Northwestern University

## Abstract

### Objective

In 2012, recommendations for universal tetanus toxoid, reduced diphtheria toxoid, and acellular pertussis (Tdap) vaccination during pregnancy were released. Our objective was to determine if Tdap, influenza, and pneumococcal vaccine uptake during pregnancy changed after the release of the guidelines, and identify factors associated with receiving the Tdap and influenza vaccine after 2012.

### Methods

We conducted a retrospective cohort study on pregnant individuals who initiated prenatal care before 20 weeks' gestation between 11/2011-11/2012 ("pre-guideline") and 12/2012-12/2015 ("post-guideline"). Vaccine uptake dates were abstracted from medical records. The pre and post-guideline cohorts were compared to determine if Tdap vaccine uptake and timing improved after the new Tdap guidelines. We additionally examined influenza and pneumococcal vaccine uptake before and after guidelines. Factors associated with receipt of the Tdap and influenza vaccine during pregnancy in the post-guideline cohort were evaluated using multivariable logistic regression models.

### Results

Of 2,294 eligible individuals, 1,610 (70.2%) received care in the post-guideline cohort. Among the pre-guideline cohort, 47.4% received Tdap, whereas Tdap uptake increased to 86.1% after the guidelines (p<0.001). Similarly, receiving the Tdap vaccine between the recommended time of 27–36 weeks gestational age improved from 52.5% to 91.8% after the guidelines (p<0.001). Vaccine frequency for influenza improved significantly from 61.2% to 72.0% (p<0.001), while frequency for pneumococcus were low and unchanged. An increased number of prenatal visits was associated with receiving the Tdap and influenza vaccines during pregnancy (respective, aOR 1.09 95% CI 1.05–1.13; aOR 1.50 95% CI 1.17–1.94). Non-Hispanic Black individuals were less likely to receive both the Tdap and influenza vaccines during pregnancy compared to non-Hispanic White individuals (respective, aOR 0.51 95% CI 0.33–0.80; aOR 0.68 95% CI 0.48–0.97).

Institutional Review Board at irb@northwestern. edu.

**Funding:** LMY was supported by the NICHD K12HD050121 at the time of the study. Research reported in this publication was supported, in part, by the National Institutes of Health's National Center for Advancing Translational Sciences, grant number UL1TR001422. The content is solely the responsibility of the authors and does not necessarily represent the official views of the National Institutes of Health. The funders had no role in study design, data collection and analysis, decision to publish, or preparation of the manuscript.

**Competing interests:** The authors have declared that no competing interests exist.

## Conclusions

Receipt and timing of Tdap vaccine improved after implementation of the 2012 ACIP guidelines. Receipt of influenza vaccine uptake also improved during the study period, while uptake of the pneumococcal vaccine remained low. Significant racial disparities exist in receipt of Tdap and influenza vaccine during pregnancy.

## Introduction

Maternal vaccines are considered essential aspects of prenatal care.(1) Multiple vaccines are recommended during pregnancy to prevent both maternal and neonatal infection, including the tetanus toxoid, reduced diphtheria toxoid, and acellular pertussis (Tdap), influenza and, when indicated, pneumococcal vaccines [1]. Although recommended, uptake of maternal vaccines during pregnancy is suboptimal throughout the United States. In fact, in 2020, 61.2% of pregnant women received the influenza vaccine, 56.6% received Tdap, and only 40.3% received both vaccines [2–4].

Pertussis is a major cause of severe morbidity and mortality for neonates [3]. In 2011, the Advisory Committee on Immunization Practices (ACIP) recommended that unvaccinated pregnant individuals receive the Tdap vaccine, which was a change from their previous guidelines that recommended vaccination of unvaccinated peripartum individuals only during the postpartum period. However, due to emerging data suggesting a short-lived antibody response from the vaccine, ACIP updated these guidelines in 2012 to recommend universal Tdap vaccination among pregnant individuals between 27 to 36 weeks' gestational age, regardless of prior vaccine status, in order to provide neonates with passive immunity [1, 3, 5, 6]. The American College of Obstetricians and Gynecologists (ACOG) and other organizations supports this recommendation [1, 3].

Yet, vaccine hesitancy is a major public health problem [1, 3, 7, 8]. Pregnant individuals who decline recommended vaccines report concerns for safety and lack of counseling regarding the benefits of vaccines by their providers [9–12]. Accordingly, vaccine uptake improves when health care providers directly recommend vaccines to their patients [4, 10, 11, 13]. Although it has been shown that providers are more likely to recommend vaccines when they are aware of recommendations from organizations such as ACIP or the Centers for Disease Control, it is not known whether these guidelines actually lead to improved uptake of recommended vaccines during pregnancy [12, 14].

Understanding vaccine hesitancy and improving vaccine uptake are more important than ever, given the new availability of coronavirus-19 (COVID-19) vaccines. Thus, we aimed to evaluate whether the 2012 ACIP guidelines were associated with improved Tdap vaccine uptake and identify patient factors associated with receiving the Tdap vaccine during pregnancy. Furthermore, we sought to examine whether the change in Tdap guidelines was associated with uptake of influenza and pneumococcal vaccines during pregnancy. We hypothesized that the public health attention given to Tdap vaccines during pregnancy may have resulted in improved recognition of the importance of all vaccines, thereby improving uptake of other vaccines.

## Methods

This is a retrospective cohort study of individuals who received prenatal care and delivered at Northwestern Medicine Prentice Women's Hospital in Chicago, Illinois from November 1,

2011 to December 1, 2015. Patients who receive care at this academic medical center are racially, ethnically, and socioeconomically diverse. Obstetric care providers at this institution include obstetrics/gynecology specialists, maternal-fetal medicine subspecialists, obstetrics and gynecology residents, and fellows.

The updated ACIP guidelines were released on October 24, 2012. Prior to these guidelines, the Tdap vaccine was recommended only to pregnant or recently postpartum individuals who had not been vaccinated within the last 3 years. The 2012 updated guidelines recommended universal Tdap vaccination between 27 to 36 weeks gestational age in every pregnancy and were approved and disseminated by the American College of Physicians, American Academy of Family Physicians, ACOG, and the American College of Nurse-Midwives [3]. At this clinical site, the new guidelines were implemented via the use of electronic medical record reminders within obstetric templates and via increased stocking of Tdap vaccines to ensure appropriate availability; provider knowledge acquisition occurred via routine professional society information dissemination.

Pregnant individuals were included in this analysis if they received prenatal care and delivered at the academic faculty practices of this care center, were 18 years of age or older, and initiated their prenatal care at 20 weeks' gestational age or earlier. Individuals with multiple pregnancies during the study period were only included for the first pregnancy. Individuals were excluded if their gestational age at first prenatal care visit was unknown. We excluded individuals with late entry to prenatal care, defined as after 20 weeks' gestational age, due to inconsistent access to medical records for individuals who initiated care at a different clinic location and the possible resultant ascertainment or misclassification bias. We further excluded individuals with an egg allergy, as they are not eligible for the influenza vaccine.

All individuals were considered eligible for the Tdap and influenza vaccines. Although the influenza vaccine is only administered during the influenza season, the length of the influenza season and prolonged availability of the vaccine (September to May) in this locality means that all pregnancies reaching viability will have overlapped with influenza season. Eligibility for the pneumococcal vaccine was assessed in accordance with the Centers for Disease Control guidelines; individuals were eligible if they had a smoking history, respiratory disease- including asthma, chronic obstructive pulmonary disease, or sleep apnea-, or diabetes (type 1 or type 2 diabetes mellitus) [15]. Although human immunodeficiency virus (HIV) is an additional indication for the pneumococcal vaccine, this cohort consisted of all HIV seronegative individuals, as individuals living with HIV receive care in a different obstetric practice. A binary variable was created for exposure status; individuals who initiated prenatal care before November 1, 2012 were grouped in the "pre-guideline" cohort, while individuals who initiated prenatal care after November 1, 2012 were grouped in the "post-guideline" cohort.

Demographic and clinical information were extracted from electronic medical records. Maternal race and ethnicity (non-Hispanic White; NHW, non-Hispanic Black; NHB, Hispanic, Asian, other or unreported) are self-reported in the outpatient or inpatient setting and collected in the medical record. Other characteristics evaluated included age, body mass index at delivery, primary payer status (commercial insurance, public insurance or uninsured), primary language (English, Spanish, other), parity, number of prenatal visits, smoking history, respiratory disease, diabetes, and gestational age at delivery ($<27$ weeks, 27–37 weeks, $>37$ weeks). Initiating prenatal care in the first trimester was defined as having a prenatal visit prior to fourteen weeks' gestational age.

Individuals in the pre-guidelines cohort were compared to those in the post-guidelines cohort using chi-squared and t-tests, as appropriate. The primary outcome was receipt of Tdap vaccine at any time point during pregnancy or up to six-weeks postpartum. Next, we evaluated the proportion of individuals who received Tdap vaccine between 26–37 weeks' gestational

age. Secondary outcomes included receipt of influenza and pneumococcal vaccines at any point during pregnancy. Multivariable models were constructed to estimate the odds of receiving each vaccine among eligible individuals in the pre-guidelines cohort compared to those in the post-guidelines cohort, accounting for potential confounders determined by significance (p-value <0.05) on bivariable analysis. Finally, in the post-guidelines cohort, we sought to identify factors associated with receiving the Tdap vaccine between 26–37 weeks' gestational age (versus not receiving it at all) via multivariable models accounting for covariates which were significant on bivariable analysis. *A priori*, we repeated the analysis for the outcome of receiving the influenza vaccine at any time point during pregnancy.

A p-value of <0.05 was used to define statistical significance and all tests were two-tailed. Statistical analyses were performed with RStudio (Boston, MA). This study was reviewed and approved by the Northwestern University Institutional Review Board. The Northwestern University Institutional Review Board granted a waiver of consent for this analysis. Data were provided by the Northwestern Medicine Electronic Data Warehouse and were anonymized prior to analysis.

## Results

Of the 2,919 individuals who received prenatal care during the study period, 2,294 were eligible for inclusion. The majority (N = 1,610; 70.2%) received their prenatal care post-guideline (Fig 1). Individuals in the post-guideline cohort were more likely to identify as NHW (57.6% vs. 53.6%, p = 0.008) and speak English as their primary language (97.4% vs. 93.7%, p<0.001) compared to individuals in the pre-guideline cohort. Individuals in the post-guideline cohort were less likely to have initiated prenatal care during the first trimester (81.2% vs. 90.1%, p<0.001) and had significantly fewer prenatal visits compared to the pre-guideline cohort

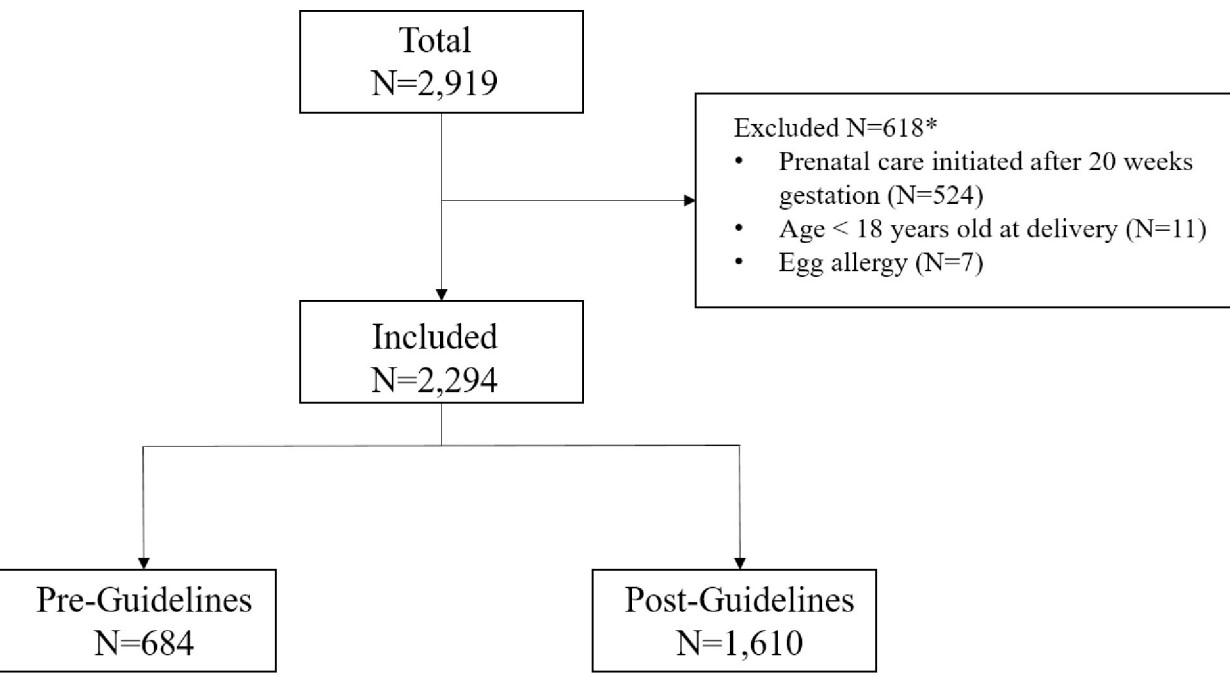

* Exclusions are not mutually exclusive.

**Fig 1. Cohort flowchart.**

**Table 1. Demographic and clinical characteristics of individuals receiving prenatal care before and after the 2012 Tdap guideline update.**

| | Post-Guidelines | Pre-Guidelines | P-value |
|---|---|---|---|
| | N = 1,610 | N = 684 | |
| **Individual characteristics** | | | |
| Age at delivery, years | 33.2 ± 4.8 | 32.8 ± 5.2 | 0.11 |
| BMI at delivery, kg/m$^2$ | 30.6 ± 5.9 | 31.0 ± 6.1 | 0.25 |
| Race and ethnicity | | | 0.008 |
| Non-Hispanic White | 924 (57.6) | 369 (53.6) | |
| Non-Hispanic Black | 199 (12.1) | 92 (13.9) | |
| Asian | 186 (10.9) | 100 (15.2) | |
| Hispanic | 119 (7.7) | 70 (9.1) | |
| Other or unreported | 182 (11.7) | 58 (8.3) | |
| Commercial insurance[a] | 1291 (80.3) | 561 (82.1) | 0.33 |
| Primary language[b] | | | |
| English | 1568 (97.4) | 640 (93.7) | <0.001 |
| Spanish | 11 (0.7) | 8 (1.2) | |
| Other | 31 (1.9) | 35 (5.1) | |
| Nulliparous[c] | 772 (50.5) | 369 (54.6) | 0.08 |
| Prenatal care initiated in first trimester | 1306 (81.2) | 612 (90.1) | <0.001 |
| Number of prenatal visits[d] | 11.2 ± 4.7 | 13.3 ± 4.0 | <0.001 |
| Smoking history (ever)[e] | 226 (14.4) | 136 (20.1) | 0.05 |
| Respiratory disease | 187 (11.6) | 74 (10.9) | 0.70 |
| Diabetes history | 64 (4.0) | 26 (3.8) | 0.959 |
| Gestational age at delivery | | | 0.12 |
| <27 weeks | 19 (1.2) | 5 (0.07) | |
| 27–37 weeks | 245 (15.2) | 84 (12.4) | |
| >37 weeks | 1342 (83.6) | 590 (86.7) | |

Data presented as N (%) or mean ± standard deviation.

BMI, body mass index; kg/m$^2$, kilograms/meters$^2$

[a] N = 2,285

[b] N = 2,287

[c] N = 2,200

[d] N = 2,280

[e] N = 2,283

(11.2 ± 4.7 vs. 13.3 ± 4.0, p<0.001). The pre- and post-guidelines cohorts did not differ by age, body mass index, insurance status, parity, smoking history, respiratory disease, diabetes, or gestational age at delivery (Table 1).

Individuals in the post-guideline cohort had a significantly higher frequency of receiving the Tdap vaccine during pregnancy or postpartum compared to individuals in the pre-guideline cohort (N = 1,385, 86.1% vs. N = 322, 47.4%, p<0.001; Fig 2). Receiving care in the post-guideline cohort was associated with an adjusted 7.37-times greater odds of receiving the Tdap vaccine during pregnancy or postpartum compared to care in the pre-guideline cohort (95% CI 5.93–9.18; Table 2). Receiving the Tdap vaccine between the recommended time of 27–36 weeks gestational age improved from 52.5% to 91.8% after the guidelines (p<0.001). Individuals in the post-guideline cohort had an adjusted 4.50-times greater odds of receiving their Tdap vaccine during the recommended window compared to those in the pre-guideline cohort (95% CI 3.54–5.72; Table 3). Similarly, receiving the influenza vaccine during pregnancy was

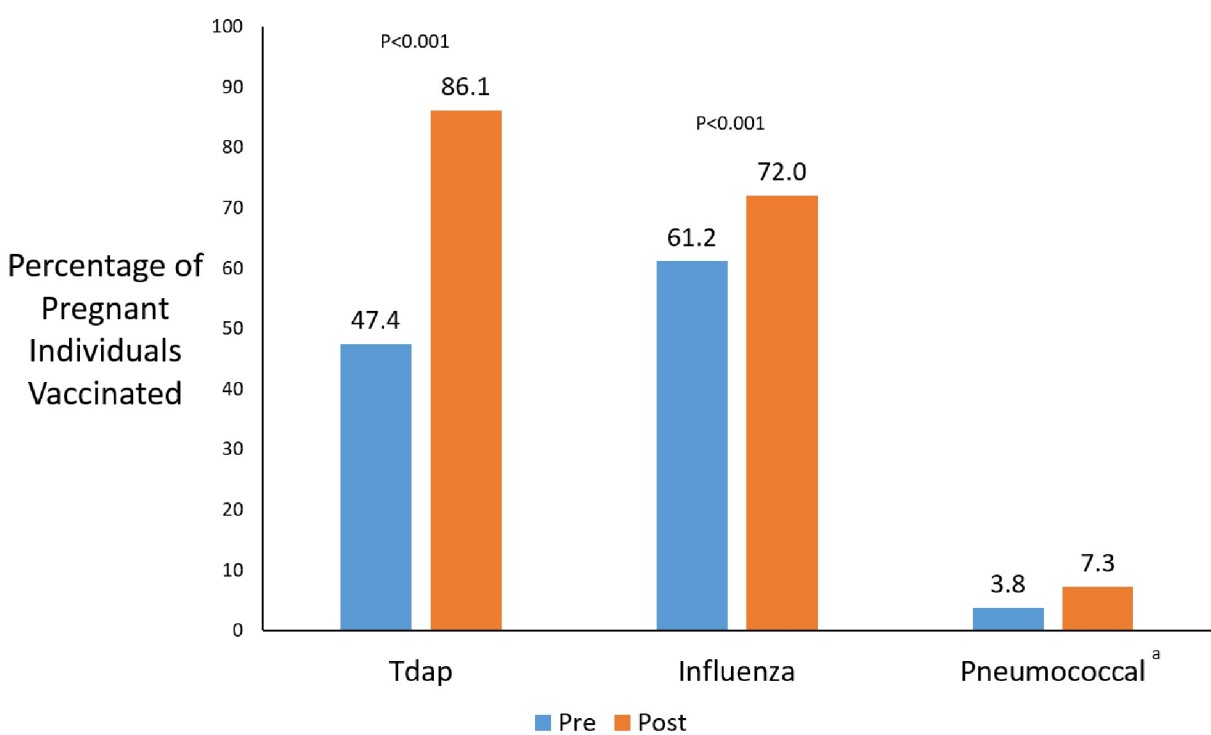

Tdap, Tetanus toxoid, reduced diphtheria toxoid and acellular pertussis.

<sup>a</sup> Analysis only among pneumococcal eligible individuals.

**Fig 2. Comparison of Tdap, influenza, and pneumococcal vaccine uptake by pre- vs. post-guidelines.**

more frequent in the post-guideline cohort than the pre-guideline cohort (N = 1,159, 72.0% vs. N = 419, 61.2%, p<0.0001; Fig 2). This finding persisted after adjustment for potential confounders; individuals in the post-guidelines cohort had an adjusted 70% increased odds of receiving the influenza vaccine during pregnancy than those in the pre-guidelines cohort (aOR 1.71, 95% CI 1.40–2.07; Table 2).

Nearly one-third (N = 675, 29.5%) of individuals were eligible for the pneumococcal vaccine. Frequency of receiving the pneumococcal vaccine during pregnancy was low throughout the study period, with only 3.8% (N = 8) of eligible individuals in the pre-guideline cohort and 7.3% (N = 32) of eligible individuals in the post-guideline cohort receiving the pneumococcal vaccine. There was no significant difference in uptake of the pneumococcal vaccine during pregnancy among eligible individuals in the post-guideline cohort compared to the pre-guideline cohort (p = 0.12; aOR 2.00, 95% CI 0.93–4.79; Table 2 and Fig 2).

Among the post-guidelines cohort, receiving the Tdap vaccine during the recommended gestational age window was significantly associated with race and ethnicity (p = 0.017), nulliparity (52.8% vs. 43.8%, p = 0.036) and increased number of prenatal visits (11.3 ± 4.7 vs. 9.5 ± 4.7, p<0.001). Individuals of NHB race and ethnicity had a 49% decreased odds of receiving the Tdap vaccine during the recommended time period compared to NHW individuals (aOR 0.51; 95% CI 0.33–0.80). Similarly, nulliparous individuals had a 43% increased odds of receiving the Tdap vaccine during the recommended time period compared to parous individuals (aOR 1.43; 95% CI 1.03–2.00). Each additional prenatal visit was associated with a 9%

**Table 2. Logistic regression of receipt of Tdap, influenza, and pneumococcal vaccine by pre- and post-guidelines cohorts.**

|  | OR (95% CI) | aOR (95% CI)[a] |
|---|---|---|
| **Vaccine** | | |
| Tdap | 6.89 (5.61–8.48) | 7.37 (5.93–9.18) |
| Tdap vaccine between 27–36 weeks gestational age | 4.40 (3.50–5.53) | 4.50 (3.54–5.72) |
| Influenza | 1.63 (1.36–1.97) | 1.70 (1.40–2.07) |
| Pneumococcal[b] | 1.98 (0.95–4.67) | 2.00 (0.93–4.79) |

OR, odds ratio; aOR, adjusted odds ratio; CI, confidence interval; Tdap, Tetanus toxoid, reduced diphtheria toxoid and acellular pertussis.

[a] Adjusted for race/ethnicity, primary language, initiation of first trimester prenatal care, and parity.

[b] Analysis only among pneumococcal eligible individuals. See methods for indications.

increased odds of receiving the Tdap vaccine during the recommended time period (aOR 1.09; 95% CI 1.05–1.13; Table 3).

Receiving the influenza vaccine during pregnancy among the post-guideline cohort was significantly associated with older age (33.4 ± 4.7 years vs. 32.8 ± 5.1 years, p = 0.017), race and

**Table 3. Factors associated with receiving the Tdap vaccine between 27–36 weeks gestational age after the 2012 ACIP recommendations.**

|  | Tdap Vaccine Between 27–36 Weeks Gestational Age | No Tdap Vaccine | P-value | aOR (95% CI)[f] |
|---|---|---|---|---|
|  | N = 1,271 | N = 176 | | |
| **Individual characteristics** | | | | |
| Age at delivery, years | 33.2 ± 4.6 | 33.6 ± 5.8 | 0.306 | --- |
| BMI at delivery, kg/m$^2$ | 30.5 ± 5.8 | 31.2 ± 5.7 | 0.124 | --- |
| Race and ethnicity | | | 0.017 | |
| Non-Hispanic White | 742 (58.4) | 84 (47.7) | | 1.00 (reference) |
| Non-Hispanic Black | 143 (11.2) | 37 (21.0) | | 0.51 (0.33–0.80) |
| Asian | 147 (11.6) | 22 (12.5) | | 0.74 (0.45–1.26) |
| Hispanic | 90 (7.1) | 14 (8.0) | | 0.76 0.42–1.50) |
| Other or unreported | 149 (11.7) | 19 (10.8) | | 0.83 (0.49–1.47) |
| Commercial insurance[a] | 1031 (88.5) | 238 (85.0) | 0.132 | --- |
| Primary language | | | 0.805[e] | --- |
| English | 1243 (97.8) | 174 (98.9) | | |
| Spanish | 8 (0.6) | 0 (0.0) | | |
| Other | 20 (1.6) | 2 (1.1) | | |
| Nulliparous[b] | 637 (52.8) | 75 (43.8) | 0.036 | 1.43 (1.03–2.00) |
| Prenatal care initiated in first trimester | 1087 (85.5) | 184 (86.0) | 0.432 | --- |
| Number of prenatal visits[c] | 11.3 ± 4.7 | 9.5 ± 4.7 | <0.001 | 1.09 (1.05–1.13) |
| Smoking history (ever)[d] | 199 (15.7) | 36 (20.4) | 0.133 | --- |
| Respiratory disease | 139 (10.9) | 25 (14.2) | 0.248 | --- |
| Diabetes | 48 (3.8) | 11 (6.2) | 0.178 | --- |

Data presented as N (%) or mean ± standard deviation.

BMI, body mass index; kg/m$^2$, kilograms/meters$^2$

[a] N = 1,445

[b] N = 1,378

[c] N = 1,443

[d] N = 1,445

[e] Fischer exact test used due to small sample size.

[f] Adjusted for race and ethnicity, parity, and number of prenatal visits.

**Table 4. Factors associated with receiving the influenza vaccine during pregnancy after the 2012 ACIP recommendations.**

|  | Influenza Vaccine | No Influenza Vaccine | P-value | aOR (95% CI)[f] |
|---|---|---|---|---|
|  | N = 1,159 | N = 449 |  |  |
| **Individual characteristics** |  |  |  |  |
| Age at delivery, years | 33.4 ± 4.7 | 32.8 ± 5.1 | 0.017 | 1.01 (0.99–1.04) |
| BMI at delivery, kg/m$^2$ | 30.5 ± 5.8 | 30.9 ± 6.0 | 0.236 | - - - |
| Race and ethnicity |  |  | 0.017 |  |
| Non-Hispanic White | 687 (59.2) | 235 (52.3) |  | 1.00 (reference) |
| Non-Hispanic Black | 128 (10.9) | 73 (16.3) |  | 0.68 (0.48–0.97) |
| Asian | 128 (11.0) | 58 (12.9) |  | 0.84 (0.59–1.21) |
| Hispanic | 83 (7.2) | 36 (8.0) |  | 0.82 (0.54–1.28) |
| Other or unreported | 135 (11.6) | 47 (10.5) |  | 0.99 (0.69–1.44) |
| Commercial insurance[a] | 951 (82.1) | 338 (75.4) | 0.003 | 1.15 (0.85–1.55) |
| Primary language |  |  | 0.957[e] | - - - |
| English | 1130 (97.5) | 437 (97.3) |  |  |
| Spanish | 8 (0.7) | 3 (0.7) |  |  |
| Other | 21 (1.8) | 9 (2.0) |  |  |
| Nulliparous[b] | 557 (50.6) | 214 (50.1) | 0.913 | - - - |
| Prenatal care initiated in first trimester | 891 (76.9) | 300 (66.8) | <0.001 | 1.01 (0.99–1.04) |
| Number of prenatal visits[c] | 11.4 (4.6) | 10.7 (5.2) | <0.001 | 1.50 (1.17–1.94) |
| Smoking history (ever)[d] | 190 (16.4) | 76 (17.0) | 0.851 | - - - |
| Respiratory disease | 140 (12.1) | 47 (10.5) | 0.017 | - - - |
| Diabetes |  |  |  | - - - |

Data presented as N (%) or mean ± standard deviation.

BMI, body mass index; kg/m$^2$, kilograms/meters$^2$

[a] N = 1,606

[b] N = 1,328

[c] N = -1,604

[d] N = 1,605

[e] Fischer exact test used due to small sample size.

[f] Adjusted for age at delivery, race and ethnicity, commercial insurance, prenatal care initiated in 1st trimester, and number of prenatal visits.

ethnicity (p = 0.017), initiating prenatal care in the first trimester (76.9% vs. 66.8%, p<0.001), and increased number of prenatal visits (11.4 ± 4.6 vs. 10.7 ± 5.2, p<0.001). Individuals of NHB race and ethnicity a 32% decreased odds of receiving the influenza vaccine during pregnancy compared to NHW individuals (aOR 0.68; 95% CI 0.48–0.97). Each additional prenatal visit was associated with a 50% increased odds of receiving the influenza vaccine during pregnancy (aOR 1.50; 95% CI 1.17–1.94; Table 4).

## Discussion

In this retrospective cohort study, we examined vaccine uptake during pregnancy before and after the introduction of the 2012 ACIP Tdap guidelines. We found that uptake of both Tdap and influenza vaccines improved after the release of these guidelines, while uptake of the pneumococcal vaccine among eligible individuals remained low. Significant disparities in maternal vaccination exist; compared to NHW individuals, NHB individuals had 49% decreased odds of receiving the Tdap vaccine during the recommended time period and 32% decreased odds of receiving the influenza vaccine during pregnancy compared to NHW individuals. An increased number of prenatal visits was associated with increased odds of receiving both

vaccines, while being nulliparous was only associated with an increased odds of receiving the Tdap vaccine.

The Tdap vaccine is highly efficacious; previous research suggests 80% protection up to one year after receiving the vaccine [16]. Receiving the Tdap vaccine during pregnancy is known to protect both the pregnant individual and newborn from pertussis. Indeed, antenatal vaccination may prevent 90% of neonatal hospitalizations and 95% of deaths due to pertussis in the first two months of life [17]. Additionally, receiving the Tdap vaccine during pregnancy has not been associated with adverse obstetric events or birth outcomes [18]. Despite this, rates of vaccine uptake are suboptimal among pregnant individuals. Previous research has suggested that providers' knowledge of professional societies' guidelines may influence prenatal vaccine advocacy. This is significant considering that pregnant individuals are more likely to receive vaccines that are recommended by their providers [10–12]. One study among obstetric providers in New York found that provider knowledge of the ACIP 2012 updated Tdap guidelines was associated with a 23.3-times greater odds of provider recommendation of the vaccine [14]. Among our cohort, Tdap vaccine uptake also improved in the post-guideline cohort, further supporting that attention to the updated guidelines may improve maternal vaccine uptake.

Multiple other antenatal vaccines are recommended for eligible recipients, including the influenza vaccine [1]. Influenza during pregnancy has been associated with multiple maternal and fetal morbidities, such as congenital anomalies, preterm birth, and low birth weight [7]. The influenza vaccine is universally recommended to pregnant individuals at any gestational age during the annual influenza season [1, 7]. Although the influenza vaccine provides maternal protection from severe morbidity, uptake during pregnancy remains suboptimal [1, 4, 11, 13, 19]. Our data demonstrate that uptake of the influenza vaccine during pregnancy significantly improved after the release of the 2012 Tdap guidelines. Although the Tdap guidelines did not alter the extant recommendations for influenza vaccine, the increased awareness of the importance of vaccines may have improved provider knowledge or patient acceptance of vaccines in general, thereby leading to an increased uptake in antenatal influenza vaccine.

Moreover, at-risk pregnant individuals should be counseled on the benefits of the pneumococcal vaccine [1]. Among our study cohort, pneumococcal vaccine uptake was low throughout the study period, consistent with prior reports of poor obstetric provider knowledge about the pneumococcal vaccine [20, 21]. A 2020 survey assessing resident knowledge on pneumococcal vaccine during pregnancy found that obstetrics and gynecology residents expressed a comfort level of 3.8/10 in regards to understanding potential candidates for the vaccine [20]. Another study on a nationally representative sample of residents found that the majority of residents demonstrated low or moderate knowledge of the indications for pneumococcal vaccine during pregnancy [21]. Areas for future work include examining whether improving provider knowledge of pneumococcal vaccine eligibility criteria results in improved patient receipt of indicated vaccines.

Significant disparities in receipt of the Tdap vaccine during the recommended gestational age of 27–36 weeks exist. NHB individuals had significantly lower odds of receiving the Tdap vaccine during the recommended time period than NHW individuals. This is consistent with prior research on racial disparities in vaccine uptake [9, 19]. However, we were unable to examine whether racial disparities in vaccine uptake widened or were ameliorated after the release of the ACIP 2012 recommendations. Whether guidelines by professional societies may alleviate racial disparities in prenatal vaccine uptake is an area for future research. In order to address racial disparities in uptake of maternal vaccines, qualitative research examining the perspectives of NHB patients on prenatal vaccines is warranted. Additionally, culturally sensitive education and intervention efforts to address this disparity should be considered.

Interestingly, number of prenatal visits, and not timing of prenatal initiation, was associated with receiving the Tdap vaccine during the recommended time period. Individuals who have more prenatal visits may subsequently spend more time with their providers, which possibly increases opportunities for discussions about vaccines and improves patient comfort with provider recommendations. Unsurprisingly, these disparities remained significant for receipt of the influenza vaccine during pregnancy, supporting the notion that disparities in prenatal vaccine uptake exist more broadly and influence general vaccine hesitancy or quality of care [8].

Our results support that guidelines by major professional societies, such as ACIP and ACOG, are associated with improvements in vaccine uptake during pregnancy. It is well documented that vaccine uptake is higher with provider recommendations and offering of vaccines in clinic [4, 10, 11, 13]. These findings remain applicable today. On January 27, 2021, the Society of Maternal Fetal Medicine and ACOG jointly released a statement stressing that the COVID-19 vaccines should not be withheld from pregnant individuals who choose to receive the vaccine [22]. With the current public health crises of COVID-19 and national vaccine hesitancy, statements and guidelines made by professional societies are likely to be important to patients and providers and may be a mechanism to support equity in receipt of vaccines. It remains imperative that providers and public health professionals are aware of vaccine recommendations in order to provide patients with the most accurate, up-to-date information [22].

Our study has several strengths. We utilized data extracted from electronic medical records, which are accurate and reliable sources of patient information. Additionally, since our study took place at a large care center, our sample size was sufficient to detect a meaningful difference in vaccine uptake. Nevertheless, our data have limitations. First, this analysis was limited to a single care center and may not be widely generalizable. Changes that may have occurred at this care center regarding the implementation of the 2012 ACIP guidelines may not have occurred at other institutions. Future research is warranted in broader settings with larger sample sizes. Secondly, we did not have data on provider recommendations, and thus are unable to determine what proportion of our cohort was provided a Tdap, influenza, or pneumococcal vaccine recommendation from their provider; future work should examine whether the identified deficits in vaccine uptake are driven by providers, patients, or both. Finally, we were unable to examine other vaccines that are recommended for eligible individuals during pregnancy, such as Hepatitis A and B, which may have similarly been affected the 2012 ACIP guidelines.

## Conclusion

In conclusion, at this particular academic institution, both frequency and timing of Tdap vaccine uptake improved after the implementation of the 2012 ACIP guidelines. Influenza vaccine uptake also improved during the study period, while rates of pneumococcal vaccine were unchanged and remained low. Significant racial disparities exist in receipt of Tdap and influenza vaccines, despite guidelines recommending their universal administration. With increasing vaccine hesitancy among the COVID-19 pandemic, it is important that providers be a reliable, patient-centered, and equitable source of vaccine-related health education in order to optimize maternal and neonatal health.

## Author Contributions

**Conceptualization:** Julia D. DiTosto, Rebecca E. Weiss, Lynn M. Yee, Nevert Badreldin.

**Data curation:** Julia D. DiTosto, Rebecca E. Weiss, Nevert Badreldin.

**Formal analysis:** Julia D. DiTosto, Lynn M. Yee, Nevert Badreldin.

**Funding acquisition:** Rebecca E. Weiss, Lynn M. Yee, Nevert Badreldin.

**Investigation:** Julia D. DiTosto, Nevert Badreldin.

**Methodology:** Julia D. DiTosto, Rebecca E. Weiss.

**Project administration:** Rebecca E. Weiss, Nevert Badreldin.

**Supervision:** Lynn M. Yee, Nevert Badreldin.

**Visualization:** Lynn M. Yee.

**Writing – original draft:** Julia D. DiTosto, Lynn M. Yee, Nevert Badreldin.

**Writing – review & editing:** Julia D. DiTosto, Rebecca E. Weiss, Lynn M. Yee, Nevert Badreldin.

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
