## [Decision Letter · Decision Letter 0]

18 Jun 2021

PONE-D-21-11490

Association of Tdap Vaccine Guidelines with Vaccine Uptake during Pregnancy

PLOS ONE

Dear Dr. Badreldin,

Thank you for submitting your manuscript to PLOS ONE. After careful consideration, we feel that it has merit but does not fully meet PLOS ONE’s publication criteria as it currently stands. Therefore, we invite you to submit a revised version of the manuscript that addresses the points raised during the review process.

We look forward to receiving your revised manuscript.

Kind regards,

Sanjay Kumar Singh Patel, Ph.D.

Academic Editor

PLOS ONE

Journal Requirements:

2. In your ethics statement in the Methods section and in the online submission form, please provide additional information about the data used in your retrospective study. Specifically, please ensure that you have discussed whether all data were fully anonymized before you accessed them and/or whether the IRB or ethics committee waived the requirement for informed consent. If patients provided informed written consent to have data from their medical records used in research, please include this information.

3. In your Methods section, please provide additional information about the participant selection method and the demographic details of your participants. Please ensure you have provided sufficient details to replicate the analyses such as:

a) the recruitment date range (month and year),

b) the institutions where patients were from,

c) a statement as to whether your sample can be considered representative of a larger population

Reviewers' comments:

Reviewer's Responses to Questions

**Comments to the Author**

1. Is the manuscript technically sound, and do the data support the conclusions?

Reviewer #1: Yes

2. Has the statistical analysis been performed appropriately and rigorously? 

Reviewer #1: Yes

3. Have the authors made all data underlying the findings in their manuscript fully available?

Reviewer #1: Yes

4. Is the manuscript presented in an intelligible fashion and written in standard English?

Reviewer #1: Yes

5. Review Comments to the Author

Reviewer #1: The manuscript entitled “Association of Tdap Vaccine Guidelines with Vaccine Uptake during Pregnancy” by DiTosto et al is conducted/performed this study to understand the vaccine uptake (Tdap, influenza, and pneumococcal vaccine) during pregnancy with respect to pre- and post- implementation of the 2012 ACIP guidelines. They observed that receipt and timing of Tdap vaccine and influenza vaccine has improved after implementation of the 2012 ACIP guidelines whereas uptake of the pneumococcal vaccine remained low. After carefully reviewing this article, this reviewer has certain suggestions that would help in more comprehensive overview of the topic.

Suggestions:

1. Since authors have studied only vaccination data of 11/2011-11/2012 (“pre-guideline”) and 12/2012-12/2015 (“post guideline”) that is form single care center only, therefore authors should include more data to their study.

2. Figure’s quality is not good for publication. Authors may improve the quality.

3. In figure 1, authors should check the total number of individuals included in their study as sum of pre-guidelines and post-guidelines individuals (684+1610= 2294) is more than total number of individuals (2288).

4. Why 2012 ACIP guidelines could not address racial disparities in vaccine uptake, authors may suggest their view to improve these guidelines further.

6. PLOS authors have the option to publish the peer review history of their article (what does this mean?). If published, this will include your full peer review and any attached files.

Reviewer #1: **Yes: **Vinay Kumar

---

## [Author Response · Author response to Decision Letter 0]

29 Jun 2021

EDITOR

Editor, Point 1:

Response: We have reviewed our reference list and determined that none of the cited papers have been retracted. 

Editor, Point 2:

Response: We have updated the manuscript and file naming in accordance with the PLOS ONE’s style requirements. 

Editor, Point 3:

In your ethics statement in the Methods section and in the online submission form, please provide additional information about the data used in your retrospective study. Specifically, please ensure that you have discussed whether all data were fully anonymized before you accessed them and/or whether the IRB or ethics committee waived the requirement for informed consent. If patients provided informed written consent to have data from their medical records used in research, please include this information.

Response: This study was approved by the Northwestern University IRB and granted a waiver of consent. Data was provided by the Northwestern Medicine Electronic Data Warehouse and included protected health information. We have updated the ethics statement in the Methods section and the online submission form accordingly. 

Editor, Point 4: 

In your Methods section, please provide additional information about the participant selection method and the demographic details of your participants. Please ensure you have provided sufficient details to replicate the analyses such as:

a) the recruitment date range (month and year),

b) the institutions where patients were from,

c) a statement as to whether your sample can be considered representative of a larger population

Response: We appreciate the opportunity to clarify. We have revised our methods section to include the requested details and are happy to make any additional changes if requested by the Editor. 

Editor, Point 5:

We note that you have indicated that data from this study are available upon request. PLOS only allows data to be available upon request if there are legal or ethical restrictions on sharing data publicly. For information on unacceptable data access restrictions, please see http://journals.plos.org/plosone/s/data-availability#loc-unacceptable-data-access-restrictions.

 Response: Thank you for the opportunity to clarify. In accordance with PLOS requirements, we are unable to share a de-identified dataset due to the potential of identifying sensitive patient information. We have clarified this in our cover letter. 

Editor, Point 6:

Response: Thank you for this information. We have updated our cover letter accordingly. 

Editor, Point 7:

PLOS requires an ORCID iD for the corresponding author in Editorial Manager on papers submitted after December 6th, 2016. Please ensure that you have an ORCID iD and that it is validated in Editorial Manager. To do this, go to ‘Update my Information’ (in the upper left-hand corner of the main menu), and click on the Fetch/Validate link next to the ORCID field. This will take you to the ORCID site and allow you to create a new iD or authenticate a pre-existing iD in Editorial Manager. Please see the following video for instructions on linking an ORCID iD to your Editorial Manager account: https://www.youtube.com/watch?v=_xcclfuvtxQ

Response: We have added the ORCID iD for Dr. Badreldin in the submission portal (0000-0002-6341-3350). 

REVIEWER 1

Reviewer 1, Point 1:

 Since authors have studied only vaccination data of 11/2011-11/2012 (“pre-guideline”) and 12/2012-12/2015 (“post guideline”) that is form single care center only, therefore authors should include more data to their study.

Response: Thank you for this feedback. We believe that our sample size is large enough to detect meaningful differences across groups. However, we agree that future research is warranted in broader settings and with larger sample sizes. We have included this in our limitations section. 

Reviewer 1, Point 2:

Figure’s quality is not good for publication. Authors may improve the quality.

Response: We apologize for this inconvenience. We have updated file type and the quality of our figures in accordance with PLOS ONE requirements.

Reviewer 1, Point 3:

In figure 1, authors should check the total number of individuals included in their study as sum of pre-guidelines and post-guidelines individuals (684+1610= 2294) is more than total number of individuals (2288). 

Response: Thank you for bringing this to our attention. We have carefully reviewed and updated the values in Figure 1.

Reviewer 1, Point 4:

Why 2012 ACIP guidelines could not address racial disparities in vaccine uptake, authors may suggest their view to improve these guidelines further.

Response: We appreciate this suggestion and agree that efforts to alleviate racial disparities in prenatal vaccine uptake are of utmost importance. We have revised our manuscript to include a discussion on the importance of acknowledging and addressing racial disparities in vaccine uptake.

---

## [Editor Report · Decision Letter 1]

6 Jul 2021

Association of Tdap Vaccine Guidelines with Vaccine Uptake during Pregnancy

PONE-D-21-11490R1

Dear Dr. Badreldin,

We’re pleased to inform you that your manuscript has been judged scientifically suitable for publication and will be formally accepted for publication once it meets all outstanding technical requirements.

Kind regards,

Sanjay Kumar Singh Patel, Ph.D.

Academic Editor

PLOS ONE

---

## [Editor Report · Acceptance letter]

9 Jul 2021

PONE-D-21-11490R1 

Association of Tdap vaccine guidelines with vaccine uptake during pregnancy 

Dear Dr. Badreldin:

I'm pleased to inform you that your manuscript has been deemed suitable for publication in PLOS ONE. Congratulations! Your manuscript is now with our production department. 

Kind regards, 

on behalf of

Dr. Sanjay Kumar Singh Patel 

Academic Editor

PLOS ONE